# Enhanced Antibacterial Activity of Clindamycin Using Molecularly Imprinted Polymer Nanoparticles Loaded with Polyurethane Nanofibrous Scaffolds for the Treatment of Acne Vulgaris

**DOI:** 10.3390/pharmaceutics16070947

**Published:** 2024-07-17

**Authors:** Sammar Fathy Elhabal, Rehab Abdelmonem, Rasha Mohamed El Nashar, Mohamed Fathi Mohamed Elrefai, Ahmed Mohsen Elsaid Hamdan, Nesreen A. Safwat, Mai S. Shoela, Fatma E. Hassan, Amira Rizk, Soad L. Kabil, Nagla Ahmed El-Nabarawy, Amal Anwar Taha, Mohamed El-Nabarawi

**Affiliations:** 1Department of Pharmaceutics and Industrial Pharmacy, Faculty of Pharmacy, Modern University for Technology and Information (MTI), Mokattam, Cairo 11571, Egypt; 2Department of Industrial Pharmacy, College of Pharmaceutical Sciences and Drug Manufacturing, Misr University for Science and Technology (MUST), 6th of October City 12566, Egypt; rehab.abdelmonem@must.edu.eg; 3Chemistry Department, Faculty of Science, Cairo University, Giza 12613, Egypt; rasha.elnashar@cu.edu.eg; 4Department of Anatomy, Histology, Physiology and Biochemistry, Faculty of Medicine, The Hashemite University, Zarqa 13133, Jordan; mohamed@hu.edu.jo; 5Department of Anatomy and Embryology, Faculty of Medicine, Ain Shams University, Cairo 11566, Egypt; 6Department of Pharmacy Practice, Faculty of Pharmacy, University of Tabuk, Tabuk 71491, Saudi Arabia; 7Department of Microbiology and Immunology, Faculty of Pharmacy, Modern University for Technology and Information (MTI), Mokattam, Cairo 11571, Egypt; nesreen.safwat@pharm.mti.edu.eg; 8Department of Clinical Pharmacology, Faculty of Medicine, Alexandria University, Alexandria 21526, Egypt; mai.shoala@alexmed.edu.eg; 9Medical Physiology Department, Faculty of Medicine, Cairo University, Giza 11562, Egypt; fatma.e.elsayed@kasralainy.edu.eg; 10General Medicine Practice Program, Department of Physiology, Batterjee Medical College, Jeddah 21442, Saudi Arabia; 11Food Science and Technology Department, Faculty of Agricultural, Tanta University, Tanta City 31527, Egypt; amira.rizk@agr.tanta.edu.eg; 12Department of Clinical Pharmacology, Faculty of Medicine, Zagazig University, Zagazig 44519, Egypt; 13National Egyptian Center of Environmental & Toxicological Research (NECTR), Faculty of Medicine, Cairo University, Cairo 11562, Egypt; n.a.nabarawy@gmail.com; 14Department of Pharmaceutics, College of Pharmaceutical Sciences and Drug Manufacturing, Misr University for Science and Technology (MUST), 6th of October City 12566, Egypt; amal.anwar@must.edu.eg; 15Department of Pharmaceutics and Industrial Pharmacy, Faculty of Pharmacy, Cairo University, Cairo 11562, Egypt; mohamed.elnabarawi@pharma.cu.edu.eg

**Keywords:** molecularly imprinted polymer, polyurethane nanofiber, acne, antimicrobial, clindamycin, inflammation, *Staphylococcus aureus*

## Abstract

Acne vulgaris, a prevalent skin condition, arises from an imbalance in skin flora, fostering bacterial overgrowth. Addressing this issue, clindamycin molecularly imprinted polymeric nanoparticles (Clin-MIP) loaded onto polyurethane nanofiber scaffolds were developed for acne treatment. Clin-MIP was synthesized via precipitation polymerization using methacrylic acid (MAA), ethylene glycol dimethacrylate (EGDMA), and azoisobutyronitrile (AIBN) as functional monomers, crosslinkers, and free-radical initiators, respectively. MIP characterization utilized Fourier-transform infrared spectroscopy (FTIR) and transmission electron microscopy (TEM) before being incorporated into polyurethane nanofibers through electrospinning. Further analysis involved FTIR, scanning electron microscopy (SEM), in vitro release studies, and an ex vivo study. Clin-MIP showed strong antibacterial activity against *S. aureus*, with inhibitory concentration (MIC) and minimum bactericidal concentration (MBC) values of 0.39 and 6.25 μg/mL, respectively. It significantly dropped the bacterial count from 1 × 10^8^ to 39 × 10^1^ CFU/mL in vivo and has bactericidal activity within 180 min of incubation in vitro. The pharmacodynamic and histopathology studies revealed a significant decrease in infected animal skin inflammation, epidermal hypertrophy, and congestion upon treatment with Clin-MIP polyurethane nanofiber and reduced pro-inflammatory cytokines (NLRP3, TNF-α, IL-1β, and IL-6) conducive to acne healing. Consequently, the recently created Clin-MIP polyurethane nanofibrous scaffold. This innovative approach offers insight into creating materials with several uses for treating infectious wounds caused by acne.

## 1. Introduction

The human body’s largest organ is the skin (1.8 m^2^); it controls body temperature, protects important organs from harm, and more. Microbiome imbalances often cause skin diseases such as acne vulgaris [1,2]. Acne appears as comedones, papules, pustules, and cysts. Many factors that cause acne and other acne vulgaris bacteria have been studied recently. The pilosebaceous unit is home to lipophilic yeasts (*Pityrosporum* species), anaerobic diphtheroids (*Cutibacterium granulosum* and *Cutibacterium acnes*), and gram-positive, coagulase-negative *cocci.* The main lesions of acne vulgaris, comedones, frequently form inside sebaceous follicles. The comedone microbiome usually has a composition similar to that of the surrounding sebaceous follicles. The etiology of acne is complex, involving inflammation and skin microorganisms. Unusual follicular hyperkeratinization, increased sebum production, acne colonization, and an inflammatory cascade cause acne, which is reported to be the cause. Acne can lower self-esteem and attitudes in teens and adults [3,4].

There are numerous approaches for treating acne vulgaris nowadays, of which topical antibiotics such as clindamycin, erythromycin, metronidazole, retinoids, or hormones are utilized to treat acne vulgaris. Chemical peeling agents, such as salicylic acid, glycolic acid, and lactic acid, are also used for treatment, as they can help reduce acne lesions and improve overall skin texture. Although acne pharmacotherapies work well, they can cause possible side effects, such as behavioral disorders, antibiotic resistance, and dry, irritated skin [5]. 

Clindamycin is a tiny antibacterial chemical agent that works well against various pathogens, including certain protozoa, most anaerobic bacteria, *staphylococci*, *streptococci*, and *pneumococci*. The bacteriostatic antimicrobial drug clindamycin inhibits ribosome function in 50s by inhibiting bacterial protein production; clindamycin is the drug of choice for treating infections of the skin and soft tissues [6]. Infections of the skin and soft tissues are treated with clindamycin. It is typically applied topically for antimicrobial purposes. Although clindamycin belongs to the Biopharmaceutics Classification System (BCS III) and is nearly insoluble in water, its salt form, clindamycin hydrochloride (HCl), is soluble in water and has poor permeability. Owing medications cannot naturally produce sustained release patterns, clindamycin HCl was chosen and employed in this investigation as a model drug of a tiny hydrophilic antibacterial compound [7].

Molecularly imprinted polymers (MIPs) represent a class of artificially designed and synthesized polymers that use templates to recognize and upload macromolecules. The template molecule(s) and functional monomer(s) complex are polymerized in the presence of excess cross-linking agents that help in creating the recognition cavities to be used for the designed purpose; MIPs can precisely rebind the template after its removal or release based on the required application, whether it aims at selective uptake of the template as in case of solid phase extraction, chromatographic applications, electrochemical detection, or sustained release of such loaded templates in the case of drug delivery applications [8]. MIPs have low cost, ease of production, stability during storage, activity retention across repeated procedures, high mechanical strength, strong thermal and chemical stability, and applicability in aggressive media. The template and cross-linkers used in molecular imprinting impact the polymer’s binding affinity and specificity. 

Different approaches are used in molecular imprinting, the most common of which is bulk polymerization, which is reported to be the most straightforward and most applied technique despite its common drawbacks, including tedious washing and grinding steps that might negatively affect the designed recognition sites. Precipitation polymerization represents an excellent approach for nanosized molecular imprints. However, the number of solvents used in the synthesis is almost 3–5 times that used in its bulk polymerization, and the particles produced are of more regular size with no need for grinding or sieving [9]. 

Electrospinning is a simple and adaptable method for making submicron- to nanometer-sized ultrafine fibers. This approach produces fibers with high porosities, small pore sizes, and large surface-to-volume ratios because of their desirable properties. Electrospun nanofibers are employed in filtration, affinity membranes, biological sciences, and optical sensors [10].

Polymeric nanofibers have great potential in acne treatment, antimicrobial dressings, biosensors, filtration, optoelectronics, and drug delivery. Modern techniques can be used to fabricate nanofibers from various polymers. Examples include electrospinning, centrifugal spinning, and pressured gyration. Leading nanofiber fabrication approaches include template synthesis, phase separation, assembly, and solution blowing. Electrospinning is the easiest and most efficient method for producing nano- to microscale fibers [11]. The procedure uses electric fields to create nanofibers with diverse morphologies from various polymers, composites, or mixes. Electrospun nanofibers replicate the extracellular matrix (ECM) in structure and morphology. Additionally, its high porosity aids in nutrition transport and gas penetration for developing cells. Electrospun nanofibers include a mechanical framework that supports cell adhesion and mimics the cellular milieu. Electrospinning produces long, continuous fibers while providing precise control over fiber dimensions, enabling process scale [12].

Polyurethane, a thermoplastic copolymer, reacts di- or tri-isocyanate with a polyol using a tin-based catalyst. Both long-term medical and non-medical uses use a well-known class of elastomers. With minor alterations, medical-grade polyurethane nanofibers have been thoroughly researched for various applications, such as tissue regeneration, medication administration, biosensor devices, wound healing, and protective clothes. Polyurethane nanofiber, a wound-dressing material for acne treatment, has benefits such as water insolubility, epithelization support, non-invasiveness, and optimal pore size for nutrition and gas mobilization. Polyurethane nanofibers’ porous nature enables gas and nutrient exchange but prevents germs from reaching the wound surface. The hydrophobic property of this polymer hinders skin contact, resulting in insufficient exudate adsorption and antibacterial agent release [13].

The present study aimed to develop clindamycin antibiotics topically through molecular imprinting technology; clindamycin was immobilized within the MIP via precipitation polymerization, after which the resulting particles were loaded within polyurethane nanofiber via electrospinning, followed by investigating their implementation as an effective way to treat acne bacterial infections. Nanofibers that can administer antibiotics with prolonged release. Furthermore, hydrophilic antibiotics may have a longer release period if the medication is encapsulated in a nanoparticle.

The morphology, physical and mechanical integrity, ex-vivo deposition and permeation, thermal properties, and chemical parameters of MIP and nanofibers were investigated. Furthermore, a cytotoxicity assay, inflammatory biomarker research, and *Staphylococcus aureus* testing were performed to evaluate the anti-bacterial activity. Finally, a histopathology investigation and an in vivo animal trial on acne were carried out, with the assessment criterion being the reduction of inflammatory, non-inflammatory, and total acne lesions.

## 2. Materials and Methods

### 2.1. Materials 

The following products were purchased from Sigma-Aldrich (Milan, Italy): methacrylic acid (MAA), azoisobutyronitrile (AIBN), acetone, methanol, ethylene glycol dimethacrylate (EGDMA), and polyurethane with tetrahydrofuran (THF). All the solvents were acquired from VWR in Milan, Italy, and were of reagent or HPLC quality. Medical International Ltd. (London, UK) provided the dialysis membranes (molecular weight cut-off, MWCO: 3500 Da) used in the in vitro release tests.

### 2.2. Animals

The Cairo University Faculty of Pharmacy Ethics Committee has approved in vivo animal research (acceptance code no. PI-3218). For in-vivo tests, eight-week-old male Wistar albino rats weighing 150 and 200 g were split into three groups. The animals were housed in specially designed, pathogen-free habitats with a 12-h light/dark cycle, constant humidity, and temperatures between 20 and 24 °C. They also had free reign to eat and drink. The investigation was also carried out by ARRIVE principles, the “Guide for the Care and Use of Laboratory Animals”, and Egyptian and local institutional legislation about animal care, and the supervision of professional examiners.

### 2.3. Methods 

#### 2.3.1. Synthesis of the Clindamycin Molecularly Imprinted Polymer (Clin-MIP)

Precipitation polymerization was employed to craft a specialized MIP tailored for clindamycin. Initially, 5 mL of methanol was utilized to dissolve the template molecule’s 424.98 mg (1 mmol) and 0.339 mL (4 mmol) of MAA in a 100 mL round-bottom flask. This solution underwent sonication for ten minutes to foster the formation of the template-functional monomer depolymerization complex. After adding 3.773 mL (20 mmol) of EGDMA, 2 mL of acetone, and 150 mg of AIBN, the reaction mixture underwent an additional ten minutes of nitrogen purging and sonication. Polymerization took place at 60 °C, with the flask rotating at 40 rpm. After 24 h, the resulting polymeric particles were extracted via filtration, and the polymeric particles were vacuum-dried overnight at 40 °C to be used for the electrospinning process. The amounts of clindamycin were tracked using high-performance liquid chromatography (HPLC) (Agilent 1100 series, Santa Clara, CA, USA; ACE 5 C18 HPLC column (5 μm, 150 × 4.6 mm), Scotland, with the detection wavelength set at 210 nm. HPLC analysis was conducted using an acetonitrile phosphate buffer solution (pH 2.5), 25:75 (*v*/*v*) in the mobile phase with a flow rate of 1.0 mL/min. The column temperature was set at 25 °C, and injections were made with a sample volume of 20 μL. The retention time was 3.5 min. analysis to verify its purity, ensuring the absence of other compounds, including clindamycin. Concurrently, a non-imprinted polymer (NIP) was synthesized using the same experimental procedure as the imprinted particles, except for omitting the template molecule [14].

#### 2.3.2. In-Vitro Characterization of Clindamycin Molecularly Imprinted Polymer (Clin-MIP)

##### Particle Size and Transmission Electron Microscopy (TEM)

Using laser light scattering, the particle size, polydispersity index, and zeta potential were measured (Malvern Zeta Sizer ZS, Malvern, Worcestershire, UK). A transmission electron microscope (TEM) (JEM-1230, Joel, Tokyo, Japan) was employed to explore the structure of the MIP containing clindamycin in the formulation. The samples were positioned on a carbon-coated grid, treated with a 1% phosphotungstic acid solution for staining, and left to air dry at room temperature before observation [1]. 

##### Fourier Transformer Infrared Spectroscopy (FTIR) 

A Fourier transform infrared spectrometer (Perkin-Elmer spectrum 100) was used to investigate the interactions between clindamycin, NIP, and MIP within the formulation [15]. 

##### Discussion of Differential Scanning Calorimetry (DSC) Analysis

The thermal characteristics of clindamycin, NIP, and MIP were assessed using a Mettler DSC 823 (Mettler Toledo, Greifensee, Switzerland) and a Julabo thermocryostate model FT100Y (Julabo Labortechnik GmbH, Seelbach, Germany). The Mettler Star program (version 9. x) was used for data analysis. Indium was utilized for instrument calibration. Samples were scanned at 10 °C/min in a 20–300 °C temperature range [15].

##### Binding Studies

Static Equilibrium Adsorption Experiments

Binding studies were carried out to investigate both the recognition properties and the selectivity of synthesized MIPs. The template was completely removed using polar solvents such as an acetic acid-methanol mixture (1:9 *v*/*v*) for 48 h, followed by methanol alone for another 48 h. i.e., non-covalent interactions were involved in the polymerization process. The imprinting efficiency of the polymer was tested through incubation with 10 mg of each of the Clin-MIP and NIP in 2 mL of Clin (0.25, 0.5, 0.75, 1.0, 1.25, and 1.5 µM) for 1 h with constant shaking. Each sample was centrifuged at 9000 rpm for 10 min, and the Clin concentration was determined by HPLC analysis using the equation obtained from the calibration curve of the drug. In order to investigate the selectivity of MIP, non-competitive binding studies were also carried out in the presence of Lincomycin (Lin), which is a structural homologue of Clin, following the same experimental procedure. The binding experiments were repeated in triplicate.

Kinetic Adsorption Study

The adsorption kinetics of the produced particles were investigated using the following procedure: A total of 10 mg of polymeric beads were combined with 2 mL of a standard solution of the substance under test (Clin) with a concentration of 1 µM in a mixture of acetic acid and methanol (1:9 *v*/*v*). The drug concentration was assessed by HPLC analysis at various incubation durations (1–24 h), followed by centrifugation of the samples at 9000 rpm for 10 min. The concentration of Clin was measured at 210 nm using the equation derived from the calibration curve of the medicinal drug. The experiments were conducted three times.

#### 2.3.3. Fabrication of Composite Polyurethane Nanofiber Dressings by Electrospinning 

Unwashed Clin-MIP was encapsulated in nanofiber using an electrospinning assembly machine (Royal HD 30, Chennai, India), and clindamycin-containing MIP powder was co-electrospun with 14 wt.% polyurethane to create nanofibers on a molecular level. Using a 500-rpm magnetic stirrer (REMI 1MLH), the polyurethane polymer was continuously mixed with 70 parts tetrahydrofuran (THF) during the night. First, various doses of Clin-MIP (0, 0.5, 1, 2, and 4% in DMF) were dispersed with the polyurethane polymer in THF. Three disposable syringes (Hindustan Syringes and Medical Devices Ltd., Faridabad, Haryana, India) with an internal needle diameter of 0.90 mm (19G) were filled with the obtained dispersions. The made-up solutions were then poured into ten-milliliter syringes and put within the electrospinning device holders. The fibers were pointed toward the depositing electrode at a 115° angle concerning the horizontal plane. A second copper loop linked the tips of the needles on the central syringe to a positive voltage source to guarantee an even charge distribution. A steady voltage of 16 kV, a tip-to-collector distance of 15 cm, a flow rate of 0.7 mL/h, a temperature of 27 °C, a relative humidity of 58%, and a drum collector speed of 140 rpm were all maintained during the electrospinning process [9]. Finally, the nanofibers were collected on aluminum foil wrapped over the metallic drum collector with an external diameter of 5 cm. Overall, 30 mL of the polymeric solution was deposited on the aluminum foil, vacuum sealed, and stored for further study. 

#### 2.3.4. In-Vitro Characterization of Clin-MIP Polyurethane Nanofiber (CMPN)

##### Consistency in Thickness and Weight

We measured the thickness of the inserts at various points on the nanofibrous mats using the Tork Craft Digital Micrometer 0–25 mm (ME30025). To ensure uniformity, we cut and weighed nanofiber pieces of equal size using digital scales for each parameter. The mean and standard deviation were then calculated and presented (mean ± SD, *n* = 3) [12].

##### Evaluation of Flexibility and Strength

To assess flexibility, the pieces were manually folded at a 180-degree angle in the center until they tore after being cut into 2 × 2 cm^2^ pieces. The folding endurance was determined by the amount of stress the fiber could withstand before rupturing. An STM-5 testing apparatus (Santam, Iran) was used to determine the skin implants’ tensile strength. The top mobile grasp of the machine applied upward force at a rate of 1 mm/min until the rectangular nanofiber samples (1.5–2 cm^2^) ruptured. Three duplicate measurements of the maximum stress withstood, the percentage elongation at the breakpoint, and the break time were made, and the results were given as means [16,17].

##### Fourier-Transform Infrared Spectroscopy (FTIR)

Using a Shimadzu IR PRESTIGE-21 (Japan), we examined the FTIR spectra of polyurethan nanofiber and Clin-MIP polyurethane nanofiber samples. Samples (*n* = 3) were compressed into analytical tablets using a manual compressing machine set to 9 tons of pressure after being ground with KBr powder. The spectrometer generated FTIR spectra within the 400–4000 cm^−1^ wavenumber range [18].

##### Morphology Characterization 

The shape of the nanofibrous scaffold was evaluated by SEM analysis. A Hitachi SU3500 SEM vacuum chamber was used to examine samples coated in gold to examine the surface characteristics of the compositions. 20 to 30 kV was the accelerating voltage [19].

##### Assessment of In-Vitro Drug Release

By employing the dialysis method, phosphate buffer solutions with a pH of 7.4 were used to investigate the release pattern of the Clin, Clin-MIP, and Clin-MIP-Polyurethane nanofibers in vitro. The nanofibers with a molecular weight cutoff of 12,000 Da were placed into dialysis bags and submerged in 100 milliliters of the corresponding buffer solutions. The solutions were gently stirred for 24 h at 200 rpm while kept at room temperature. Five milliliters of the supernatant solution were removed and replaced at predetermined intervals with an equivalent volume of brand-new buffer solution. We computed the medication release percentage using Equation (1) [20].
Drug Release %= AR/AC × 100 (1)

In this case, AC is the total amount of clindamycin that was administered to the nanofiber, and AR is the amount of clindamycin that was absorbed by the system, determine the concentration of Clin in phosphate buffer (pH 7.4) and establish the HPLC assay’s reliability, it was performed five times a day for five days in a row under the same conditions, covering a concentration range of 0.05–100 µg/mL [21].

### 2.4. Ex Vivo Skin Permeation Study

Using Clin-MIP-Polyurethane nanofiber, we performed a skin permeation study to examine medication penetration through the skin utilizing Franz diffusion cells. The Cairo University Faculty of Pharmacy Ethical Committee’s (acceptance code no. PI-3218) set of ethical principles was scrupulously followed. For the investigation, the skin of a newly shaven rat’s abdomen was used. The stratum corneum was positioned toward the donor compartment by attaching the skin to two vertical Franz diffusion cells, creating a 2.75 cm^2^ permeation region. To replicate blood pH, 100 mL of pH 7.4 phosphate buffer was added to the receptor compartment and shaken at 100 rpm. The donor cells were supplemented with 32 mg of Clin-MIP-Polyurethane nanofiber or 10 mg of clindamycin hydrochloride in an exact measurement. Concurrently, the reactor compartment holding 30 milliliters of phosphate buffered saline solution (PBS) with a pH of 7.4 was maintained at 37 ± 1 °C with magnetic stirring at 100 revolutions per minute. After six hours of removing 0.5 mL of the permeation media, the receptor cell was filled with fresh media every hour. After the samples were filtered through a 0.45-mm membrane, they were examined using the previously mentioned HPLC spectroscopy procedure. The average of the three runs was given as the result [1,22]. By charting the drug’s passage through the skin over time, Equation (2) might be used to determine the permeability coefficient (cm/h).
Kp = Jss/C_O_
(2)
where Jss (steady-state flux) is the slope of the linear section (mg/cm^2^/h) and C_O_ is the initial drug concentration (mg/cm^3^). After the experiment, any remaining medication was quickly eliminated from the skin tissues by removing them from the diffusion cell and immersing them in pure water for ten seconds. To extract the drug that had been deposited, the tissues were next chopped into tiny pieces and sonicated for 30 min in a bath that contained 5 mL of methanol. Lastly, the samples were examined using the HPLC assay that was previously explained. 

### 2.5. In Vitro Cell Toxicity Assessment

Cell toxicity experiments conducted on human skin epithelial cells (HSE-2) assessed the formulation’s safety. The ATCC PCS-201–012 cell line was procured from Vacsera Egypt. Nanofiber pieces weighing 5, 2.5, and 1.25 mg were immersed in 10 milliliters of Eagle’s modified medium with Dulbecco. Direct cell contact was allowed after 48 h following the isolation of the supernatant. HSE-2 cells were cultured in 96-well plates using DMEM/F-12 medium supplemented with 10% FBS, 100 units/mL penicillin, and 100 μg/mL streptomycin for 24 h at 37 °C with 5% CO_2_. One row served as a control with no formulations, while each of the remaining rows received supernatants from various formulation concentrations. DMSO was administered in a row as a control to inhibit cell growth. After adding trypsin, the plates were washed with PBS, and the culture medium was discarded the next day. Samples were incubated in wells with 10 μL of MTT solution (5 mg/mL) for 4 h to assess the viability of living cells [23]. The absorbance of the samples was measured at 570 nm, and the average of three tests was calculated. Cell viability was determined using Equation (3).
Cell viability (%) = (sample absorbance)/(control absorbance) ×100 (3)

### 2.6. Determination of Antimicrobial Activity

#### 2.6.1. Agar Well Diffusion Assay

Using the agar well diffusion assay, the antimicrobial activity of clindamycin and clindamycin nanofiber was evaluated against standard bacteria that may be present in the skin flora, including *Streptococcus mutans* ATCC 25175, *Staphylococcus aureus* ATCC 6538, *Pseudomonas aeruginosa* ATCC90274, *K. pneumoniae* ATCC 13883, and *Proteus vulgaris* ATCC13315. The Giza, Egypt-based Egyptian Company for Biological Products and Vaccines (VACSERA) supplied the standard bacterial strains used in the skin profile. Using Muller-Hinton agar (MHA) technology, 1.8 × 10^8^ CFU/mL (0.5 OD600) of the studied strains of bacteria were seeded onto agar plates from Oxoid, Hampshire, UK. 100 µL of a solution containing 10 mg/mL of clindamycin and its nanofiber component dissolved in 5% DMSO was added to the 6 mm diameter wells punched in the agar plates. Additionally, the examined plates were incubated at 37 °C for 24 h, while gentamycin (Sigma, St. Louis, MO, USA) was used as a positive control. According to CLSI guidelines, the inhibitory zones produced after incubation were quantified in millimeters (NCCLS, 2010) [24].

#### 2.6.2. Bacterial Proliferation Inhibition Analysis

A two-fold broth dilution method was used to determine the minimum bactericidal concentration (MBC) and minimum inhibitory concentration (MIC) of Clin and Clin nanofiber against the studied bacterial strains [4]. A 96-well microplate filled with Mullar Hinton broth was diluted with Clin. and Clin_nanofiber, resulting in final concentrations of 0.78, 1.56, 3.13, 6.25, 12.5, 25, and 50 μM. Each strain of bacteria under study was injected in equal amounts (10^6^ CFU/mL). Using a microplate reader (Infinite M1000 Pro, Tecan Company, Männedorf, Switzerland), the absorbance of bacterial growth was measured at 600 nm following a 24-h incubation period at 37 °C. The MIC values showed the lowest concentration needed to halt growth that could be seen. The MIC test was used to calculate the lowest bactericidal concentration (MBC).

#### 2.6.3. Bacterial Killing Kinetic Assay

According to Wu et al., 2021, the bacterial killing kinetics of Clin_nano-formula against acne-inducing *Staphylococcus aureus* ATCC 6538 were conducted [4]. Fresh Muller Hinton broth was combined with an equal volume of bacteria and clin nanofiber at a final concentration of 1× MIC. At different intervals (0, 15, 30, 45, 60, 90, 120, and 180 min), duplicate samples containing 0.1 milliliters were taken out and spread out on agar plates. The viable colonies were counted following a 24 h incubation period at 37 °C. Additionally, a blank containing a 1× MIC value of clindamycin was used as the positive control.

### 2.7. In-Vivo Study

To evaluate the anti-acne effectiveness of Clin suspension, Clin-MIP, and Clin-MIP polyurethane nanofiber, we employed a rat model for acne in vivo. Rats demonstrate analogous physiological processes of acne formation, making this model highly advantageous for assessing the efficacy of the prepared biotherapeutic polymer scaffold on the skin. Male Wistar albino rats weighing 150–200 g was utilized in the study. Chronic skin inflammation was induced using *Staphylococcus aureus* (ATCC 6538) through intradermal injection of bacteria into the rats’ ears. After shaving and pre-fixing with double-sided tape, fifty microliters of bacterial cultures were intradermally injected into the backs of the rats’ left and right ears. The untreated left ear side was a control, while the right ear was treated with formulations. Subsequently, the respective animal groups received consistent volumes of topically applied Clin suspension, Clin-MIP, and Clin-MIP polyurethane nanofiber for four consecutive days. The rats were divided into three groups (*n* = 8). All ears on the left side acted as the control group, while the right ear side was treated as follows: Group I received Clin suspension, Group II was treated with Clin-MIP, and Group III received Clin-MIP polyurethane nanofiber scaffold [25].

### 2.8. In Vivo Evaluation of the Prepared Clin-MIP Polyurethane Nanofiber (CMPN) 

#### 2.8.1. Induction of an Inflammatory Acne Model Using the Viable Count of Infected Rat Skin

To evaluate the in vivo anti-acne efficacy of Clin, Clin-MIP, and Clin-MIP polyurethane nanofiber, the reduction in bacterial counts was investigated after the full treatment period in infected rats. Inflammatory acne was induced in the rats by intradermally injecting viable *S. aureus* into the right-side ears. Skin pieces from both treated and untreated (control) ears were excised, and punch biopsies were homogenized in 20 μL of sterile saline. The bacterial suspensions of *S. aureus* were then serially diluted (1:10^2^–1:10^8^), and 0.1 mL of each dilution was plated on a Muller-Hinton agar plate. After incubation for 24 h at 37 °C, the colony-forming units (CFUs) of *S. aureus* were counted. Before sacrifice, all animals were intraperitoneally sedated with a 10% ketamine and 2% xylazine (1:1) solution at a dosage of 0.1 mL per 100 g of body weight.

#### 2.8.2. Assessment of Inflammatory Biomarkers 

Pentobarbital sodium (200 mg/kg, IP) was used to anesthetize each rat twenty-four hours following the prior treatment, and blood samples were taken via the retro-orbital sinus. Then, according to the manufacturer’s recommendations, serum was tested for tumor necrosis factor-alpha (TNF-α), leucine-rich repeat and pyrin domain-containing protein 3 (NLRP3), nucleotide-binding oligomerization domain, IL-1β (interleukin-1 beta), and interleukin-1 (IL-6). The enzyme-linked immunosorbent test (ELISA) kit that was used was supplied by MyBioSource, San Diego, CA, USA.

### 2.9. Histopathological Examination

Skin specimens were supplied by the rat dorsum, which acted as a fixed area for all the rats. Following a 24 h fixation in a 10% buffered formalin solution, the specimens were cut off, washed in xylol, and dried using escalating ethanol concentrations before being embedded in paraffin. Serial skin segments were sliced and layered on glass slides to a thickness of five to seven micrometers. The tissue sections were prepared, deparaffinized with xylol, and stained with hematoxylin and eosin (H&E) before being examined histologically under an electric light microscope [26].

#### Assessment of Ear Thickness and Anti-Inflammatory Potential on the Skin’s Surface 

Using the histopathological findings, scores were assessed as follows: − (no abnormality), + (mild), and ++ (moderate) for alterations. Signs of inflammation were observed, including redness, pustules, lesions, and comedones. The thickness of the degree was also used to measure the length of the rats’ ears [27].

MAA, EGDMA, and AIBN were used as functional monomers, crosslinkers, and radical initiators, respectively.

## 3. Results and Discussion

### 3.1. Synthesis of Clindamycin Molecularly Imprinted Polymers

Functional monomers such as MAA and EGDMA as crosslinkers and AIBN as radical initiators play pivotal roles as crosslinkers and radical initiators in the non-covalent imprinting process. During precipitation polymerization, these components generate oligomer radicals that crosslink to form nuclei, gradually accumulating monomers and eventually reaching a critical mass where they precipitate, forming polymeric beads. In addition to being straightforward to prepare, compatible with high crosslinker concentrations, free of stabilizers and additives, and able to use aprotic solvents as porogens to promote non-covalent interactions between the template and monomer, the non-covalent imprinting process has several benefits. The non-covalent approach is to create clindamycin-imprinted particles because it is a commonly used technique for creating MIPs. During polymerization and rebinding, this technique facilitates the creation of reversible non-covalent contacts (ion pairs, hydrogen bonds, van der Waals forces, and dipole-dipole interactions) between the target molecule and functional monomers. Methacrylic acid’s capacity to create reversible hydrogen bonds with the drug molecule is why it was selected as a functional monomer. Meanwhile, EGDMA enhanced the polymer’s mechanical strength and preserved the monomers’ functions surrounding the template molecule and recognition cavity architecture, which helps in the release of clindamycin from the polymeric matrix. MAA and EGDMA, being biocompatible polymers, have a well-established track record in pharmaceutical dosage form, drug delivery, nanotechnology, and biomedical polymer synthesis [28,29]. 

### 3.2. In-Vitro Characterization of Clindamycin Molecularly Imprinted Polymer (Clin-MIP)

#### 3.2.1. Particle Size and Transmission Electron Microscopy (TEM)

DLS was used to analyze particle size and polydispersity index, as shown in Table 1. The decreased size of NIP nanoparticles compared to MIP nanoparticles may be due to clindamycin and hydrogen interaction in the reaction medium, MAA functional monomer. The hydrogen interaction affects particle nucleation and growth, making MIP nanoparticles larger than NIP ones. Additionally, decreasing cross-linker or increasing functional monomer concentration results in smaller particles.

TEM imaging was used to characterize the enhanced clin-MIP formulae and MIP without clindamycin, as shown in Figure 1a. Photos showed A non-uniform distribution of monomers and crosslinkers polymerizes forming flower-shaped nanoparticles with a crystallite size of around 25–45 nm. Polymeric flower beads form when nuclei expand and grab monomers and oligomers until they reach critical nanoparticles. The in vitro release profile showed that irregular, flower-shaped nanoparticles released clindamycin well. The particle shape affects particle surface-medium interactions, which is critical for the regulated release of encapsulated contents, according to the literature [30,31].

#### 3.2.2. Fourier Transformer Infrared Spectroscopy (FTIR)

Figure 1bI shows the FTIR data that provided evidence for the production of Clin and Clin-MIP. At frequencies of 3306 cm^−1^ (O-H stretching vibrations), 2891 cm^−1^ (C-H stretching vibrations), 990 cm^−1^, and 1900 cm^−1^ (C-O stretch), a substantial absorption of Clin or Clin-MIP was observed in the FTIR spectra. MAA, EGDMA, and AIBN were used as functional monomers, crosslinkers, and radical initiators, respectively, as revealed by a characteristic band at 1190 cm^−1^. The cross-linking group’s C=O stretching vibrations were identified as the source of this band. Carboxylic acid moieties caused the absorption at 1724 cm^−1^ [6,28]. Melatonin did not interact with Clin and Clin-MIP in any of the synthesized samples, but all had similar spectra with similar bands at distinctive wave numbers and relative intensities. The presence of carboxylic groups’ OH groups in the Clin and Clin-MIP was ascribed to the expansion of the absorption band in the 3300–2810 cm^−1^ region. The shape and position of the peaks of NIPs are very similar to MIP with Clin. The creation of the delivery systems required an understanding of the interactions between polymers and drugs. Thus, it was imperative to employ computational tools to examine these connections simultaneously [32].

#### 3.2.3. Discussion of Differential Scanning Calorimetry (DSC) Analysis

The DSC analysis provides valuable insights into the thermal properties and physical state. Amorphous polymers have a glass transition temperature of 60 °C, as shown by the DSC analysis of the NIP as shown in Figure 1bII. The absence of a melting peak shows the NIP has few crystalline areas. NIP thermal stability is shown by thermal deterioration at 240 °C. This shows that the NIP is mostly amorphous and stable under test conditions. Pure Clindamycin has a strong melting peak at 150 °C in the DSC thermogram, showing crystallinity. Pure, crystalline compounds have well-defined melting points. Understanding Clindamycin’s physical condition before and after polymer matrix incorporation requires this information. Clin-MIP has 150 °C on its DSC thermogram, somewhat higher than NIP (50 °C). Clindamycin may interact with the polymer matrix due to the successful creation of molecularly imprinted sites as the transition temperature increases. The Clin-MIP’s Clindamycin melting peak is absent or reduced, indicating that the molecules are thoroughly integrated into the polymer matrix, disturbing their crystalline form. This disruption indicates successful molecular imprinting, where the template molecule (Clindamycin) alters the polymer’s thermal characteristics [33]. Clin-MIP degrades at 300 °C, somewhat higher than NIP (270 °C), suggesting that Clindamycin slightly improves polymer thermal stability. Clindamycin and the polymer matrix may interact to improve heat stability and structural stability.

#### 3.2.4. Binding Studies

##### Imprinting Efficacy and Selectivity

Understanding the adsorption isotherms enables one to characterize the nonlinear and dynamic equilibrium that exists between the quantity of template in the solution and the quantity of template adsorbed on the polymeric matrix. To further optimize the utilization of molecularly imprinted polymers, isotherm data analysis is a critical step. A curve that connects the equilibrium concentration of the same analyte in solution (Ce) to the amount of an analyte adsorbed onto the polymer at equilibrium (Qe), represented in this case by the template molecule, is known as an adsorption isotherm. Based on the kind of adsorption process that takes place, the Qe/Ce relationship can change. As a result, one can obtain important information on the template polymer interaction through adsorption isotherms, which can look different depending on the model used. The amount of Clin bound to the polymers at equilibrium (Qe, mol/g) may be obtained from the binding experiments carried out, allowing us to calculate the binding capacity of MIP and NIP particles Equation (4) [34].
(4)$$\mathrm{Qe}=\frac{(Ci-Ce)\times V}{m}$$

The solution’s volume is denoted by V (L), the weight of the polymers is indicated by m (g), and the initial and equilibrium Clin concentrations are indicated by Ci and Ce (mol/L), respectively. 

By plotting Qe versus Ci, the adsorption isotherms of Clin on MIP and NIP particles were obtained (Figure 1c). The outcomes of the current research investigation have verified that the imprinted particles possess the capability to bind a greater quantity of medication in comparison to the matching non-imprinted particles. In addition, as the initial concentration of Ci in the Clin grew, the adsorption capabilities of both polymeric materials likewise increased until reaching a saturation point. Lincomycin (Lin) standard solutions at different concentrations were used to conduct non-competitive binding tests. The purpose of these tests was to investigate the cross-reactivity of MIP in the presence of a structural equivalent of Clin (Figure 1c). The data in Table 1b indicates the efficacy of the imprinting procedure and the selective binding ability of the produced imprinted particles to their target molecules, as demonstrated by the percentages of bound Clin and Lin. The binding sites present in the imprinted polymers were created during the polymerization process, which took place in the presence of the specific medicinal drug. Conversely, non-specific interactions account for the quantity of Lin that is bound to both polymeric matrices. The imprinting factor (α) and the selectivity coefficient (ε) were calculated for the synthesized polymeric materials, and the findings are presented in Table 1b. The initial parameter signifies the ratio of the analyte (either the template or its counterpart) absorbed by MIP to the analyte absorbed by NIP. The second parameter is determined by calculating the ratio of Clin to Lin adsorbed by MIP. The α and ε values provided evidence of the selective recognition capabilities of the imprinted material for the therapeutic drug, as opposed to the similar non-imprinted material. The α Clin values found for each adopted Ci are greater than 1.23, indicating that the MIP has a superior ability to bind the therapeutic drug compared to a similar non-imprinted polymer. Furthermore, the imprinted particles exhibit selectivity for Clin that is 2.52 to 3.04 times higher, as indicated by the measured ε values. To better understand the adsorption properties of the synthesized MIP and the distribution of affinity sites, the binding data was analyzed using the Scatchard model equation (5) [35].
(5)$$\frac{Qe}{Ce}=(B_{\max}-Qe)Ka$$

This model helps distinguish between homogeneous and heterogeneous sites. In the equation, Qe represents the amount of Clin bound per gram of polymeric material at equilibrium (mol/g), Ce is the drug concentration at equilibrium (mol/L), B_max_ (mM/g) is the maximum binding capacity, and K_a_ (M^−1^) is the association constant. Thus, the binding parameters K_a_ and B_max_ were determined by graphing Qe/Ce against Qe, with the slope representing K_a_ and the intercept representing B_max_. These estimates were obtained from Figure 1d and Table 1c. Figure 1d demonstrates that the relationship between Qe/Ce and Qe in MIP is not linear. Instead, it consists of two distinct straight lines with varying slopes. This suggests that the imprinted particles have heterogeneous binding cavities, which can be categorized into high-affinity and low-affinity sites. In contrast, the NIP Scatchard plot consists of a solitary line, showing the existence of only one type of binding site.

##### Adsorption Kinetics

Adsorption kinetics refers to the measurement of the rate at which adsorption occurs over time while maintaining a constant concentration of the medication. The current study investigated the kinetics of adsorption by submerging 10 mg of the polymeric particles in 2 mL of a Clin standard solution (1 µM) and measuring the drug concentration at various time intervals. The quantity of a therapeutic agent that was attached at time t (Qt, mol/g) was calculated by subtracting the initial concentration of Clin at t = 0 (Ci, mol/L) from the remaining concentration at the adsorption time t (Ct, mol/L), as described in Equation (6) [35].
(6)$$\mathrm{Qt}=(Ci-Ct)\frac{V}{M}$$

The variable V (L) represents the volume of the incubation solution, while m (g) represents the quantity of polymeric material. Figure 1e displays the adsorption kinetic curves of the synthesized imprinted and non-imprinted particles, represented by the plot of Qt against t. The process of adsorption achieved a state of fast equilibrium within the initial 10 h, followed by a slower phase until it eventually reached a state of equilibrium. At this point, a plateau phase was seen. Moreover, the adsorption capacities of non-imprinted particles were consistently lower than those of comparable imprinted particles at all time points. Following the initial hour, approximately half (50%) of the clindamycin was absorbed by MIP. The absorption levels increased gradually over time, reaching 66.69%, 78.63%, 83.73%, 90.55%, 90.80%, 90.89%, and 91.06% at the time intervals of 1, 4, 6, 8, 12, 24, 28, and 30 h, respectively. However, the quantity of clindamycin that was attached to the particles without imprinting was 30% within the first hour. This amount increased to 49.63%, 49.63%, 50.51%, 50.55%, 50.51%, 50.85%, and 50.85% at the time intervals of 10, 12, 16, 20, 24, 28, and 30 h, respectively. The collected experimental data was used to fit kinetic models of pseudo-first order and pseudo-second order to examine the kinetics of Clin adsorption on the produced MIP. The pseudo-first-order or Lagergren model is expressed by the following Equation (7) [34]: (7)$$\log(\mathrm{Qe}-\mathrm{Qt})=\log(\mathrm{Qe})-\frac{K_{1}}{2.303}t$$

The variable Qe indicates the quantity of adsorbed Clin at the point of equilibrium, while K_1_ denotes the rate constant for first-order adsorption and t represents the duration of adsorption. This model operates on the assumption that a single molecule is adsorbed within the active site of the polymer. Furthermore, it assumes that the process of drug adsorption, which is regulated by the diffusion step, falls under the category of physical adsorption. Equation (8) describes the pseudo-second-order model.
(8)$$\frac{t}{Qt}=\frac{1}{K_{2}Qe2}$$

K_2_ represents the rate constant for adsorption in a pseudo-second-order reaction. This kinetic model postulates that a single adsorbate molecule interacts with two active sites and that the adsorption process is a chemical process, which may be the step that limits the rate. The kinetic fitting data for the imprinted and non-imprinted particles are documented in Table 1d. An analysis was conducted to determine the suitability of the two kinetic models for describing the adsorption behavior of MIP and NIP. This analysis examined the correlation coefficients (R^2^) as well as the experimental and estimated binding capacity (Qe). The adsorption process exhibited pseudo-first-order kinetics for both the imprinted and non-imprinted materials, as indicated by the higher R^2^ values. Furthermore, the Qe values calculated using the pseudo-first-order kinetic model were more accurate compared to the Qe values obtained from the pseudo-second-order equation, indicating that the latter was not appropriate for the experimental conditions used. In addition, the larger MIP K_1_ value indicates that the imprinted particles have a noticeably faster adsorption rate compared to the non-imprinted particles. As observed from the results for MIP and NIP, both models fit the data well, but MIP exhibits a higher adsorption rate than NIP based on the pseudo-first-order kinetic model rate constants (K_1_).

### 3.3. Fabrication of Composite Polyurethane Nanofiber Dressings by Electrospinning 

This study investigated polyurethane nanofibers loaded with Clin-MIP and composites as antibacterial dressings for treating acne. Several trial tests were conducted with different voltages, flow rates, needle-to-collector distances, polymer weights, and Clin-MIP concentrations to determine the ideal polyurethane concentration for bead-free fibers. These screening studies discovered fibers with unusual aggregation patterns, such as beaded, distorted, or both. As mentioned in the technique section, bead-free fibers were produced with a polyurethane content of 14% (*w*/*w*) and a solvent ratio of 70:30 (THF: DMF). For electrospinning, Clin-MIP in DMF at a 4% polyurethane concentration was selected. Beyond this range, the viscosity of the polyurethane solution decreased, and the rate of dripping increased with the clindamycin ions in the clindamycin salt [36].

### 3.4. In-Vitro Characterization of Clin-MIP Polyurethane Nanofiber (CMPN)

#### 3.4.1. Thickness and Weight Uniformity

Test results for thickness and weight homogeneity are in Table 2. Nanofiber pieces with identical sizes had a mean weight of 31.54 ± 0.1 to 22.65 ± 0.2 mg. The SD was less than 0.7% of the mean weight values. Indicated formulation uniformity. A higher thickness was observed in Clin-MIP polyurethane nanofiber (0.28 ± 0.01) compared to polyurethane nanofiber (0.18 ± 0.01). The nanofibrous mat thickness was uniform throughout both formulations. The right thickness and weight of the prepared dressing make it easy to use [13,28]. 

#### 3.4.2. Flexibility and Strength Testing Pieces of Nanofiber

One popular test for assessing dressing implant flexibility is folding endurance, as shown in Table 2. Nanofibers showed a folding endurance ranging from 154 ± 5 to 174 ± 3 times. Both inserts appear strong enough to resist ripping after being placed in the skin. For optimal folding endurance, fibers should not irritate the skin or dissolve quickly in the skin. Both implants showed strong tensile strength for skin use. Table 2 data indicates that Clin-MIP polyurethane nanofiber (CMPN) has lower strength, elongation, and break time than Clin-MIP polyurethane nanofiber (CMPN). The higher inherent flexibility of Clin-MIP polyurethane nanofiber compared to polyurethane fiber may explain the higher strength of Clin-MIP polyurethane nanofiber compositions [17]. 

#### 3.4.3. Morphology Characterization of Clin-MIP Polyurethane Nanofiber 

A scanning electron microscope (SEM) measures the morphology and diameter of silk fibers. On the other hand, the SEM pictures of the Clindamycin-MIP-polyurethane nanofibers show a fibrous morphology that is comparable to that of the pristine polyurethane nanofibers. This suggests that adding molecularly imprinted polymer (MIP) did not substantially change the nanofibers’ overall structure. Nevertheless, the performance and functionality of the nanofibers may be impacted by minute alterations in the surface morphology and porosity brought about by the presence of clindamycin-MIP in the polyurethane matrix. The produced fibers had continuous and homogeneous structures. Clin-MIP polyurethane nanofiber (CMPN) had random nanofiber alignment, with a mean diameter of 411 ± 3.32 nm, as shown in Figure 2a,b. When fiber diameter increases, tensile strength and modulus improve.

Figure 2c shows Polyurethane nanofibers’ FTIR spectra show distinctive absorption bands indicative of the functional groups found in the polymer chain. The stretching vibrations of N-H and O-H groups correlate to the peaks at around 3300–3500 cm^−1^, suggesting the presence of urethane and hydroxyl functionalities. The absorption bands located about 1700–1750 cm^−1^ are ascribed to the carbonyl groups in the urethane linkage’s C=O stretching vibrations. Furthermore, the C-O stretching vibrations of the ether links in the polymer backbone correspond to maxima at about 1200–1250 cm^−1^.

Similar absorption bands corresponding to polyurethane functional groups are seen in the FTIR spectra of the clindamycin-MIP-polyurethane nanofibers, suggesting that the polymer structure is retained following the inclusion of molecularly imprinted polymer (MIP). Comparing the FTIR spectra of polyurethane nanofibers with Clindamycin-MIP-polyurethane nanofibers sheds light on the materials’ structural and chemical makeup. The presence of clindamycin-MIP may cause changes to certain functional groups or bonding interactions within the polymer matrix, even when the overall spectral characteristics stay the same. These modifications can signify the integration of clindamycin-MIP and its possible impact on the physicochemical characteristics of the nanofibers.

### 3.5. In-Vitro Drug Release 

Clindamycin release patterns from MIP and MIP-polyurethane nanofibers were measured and contrasted with those from the clindamycin solution shown in Figure 3.

All formulations in this study contain 424.98 mg clindamycin, 4 mmol MMA, 150 mg AIBN, and 20 mmol EGDMA. Anticipatedly, during the first hour of the trial, 40 ± 0.87% of the medication from the drug solution was released, and by the fourth hour of the experiment, 98 ± 2% had been released. On the other hand, the release of clindamycin could be considerably controlled (*p* < 0.05) by the nanoparticles at all time points, irrespective of their MIP—polyurethane nanofiber. The drug release characteristics of the two formulations were rather close to one another, releasing only roughly 77% and 50% of the clindamycin from Clin-MIP and Clin-MIP-Polyurethane nanofiber, respectively, after 30 h. NIP exhibits the least amount of drug release, suggesting a reduced affinity for the medication and a lack of binding sites. 30 h of release of 38 µg is significantly less than that of MIP and MIP-nanofiber. This property will enable MIP to have a rapid onset of action; this variation releases the drug more quickly than the nanofiber type and nanofiber systems, potentially extending the duration of the therapeutic impact, which would be advantageous for the dermatological treatment of acne should they concentrate within the skin’s follicular openings following topical application [20,37].

### 3.6. Ex-Vivo Skin Permeation Study

Franz diffusion cells combined with rat skin as a permeation membrane were used in ex-vivo permeation investigations. All formulations in this study contain 424.98 mg of clindamycin, 4 mmol of MMA, 150 mg of AIBN, and 20 mmol. As a permeation medium, phosphate buffer (pH 7.4) was employed. At 24 h, clindamycin from Clin-MIP exhibited less skin penetration than Clin-MIP-Polyurethane nanofiber (Figure 4; Table 2); the steady state flux and permeability coefficient are shown in Table 3. Polyurethane has been shown to reduce the drug’s crystallinity, which increases the amount of drug released from the nanofiber as concentration rises [1,21]. MAA was utilized as a monomer since it diffuses and softens the polymer matrix. In addition to creating a link with other polymer molecules that aid in creating MIP-polyurethan nanofiber, it lessens the interactions between polymer molecules, such as hydrogen bonding. An analysis was also completed on the impact of particle size on permeation. The findings demonstrated that the degree of drug penetration through the stratum corneum increased with decreasing particle size; the smaller the particle size, the higher the surface of the particles and the more intimate their contact with corneocytes. Analysis of the impact of surfactant on permeation revealed that a rise in the ratio of surfactant to lipid corresponded to an increase in medication penetration through the skin. The surfactant may facilitate medication penetration by loosening the stratum corneum’s lipid bilayers [38,39].

### 3.7. In Vitro Cell Toxicity 

Using HSE-2 cells, the safety of clindamycin loaded with MIP polyurethane nanofiber was investigated. The results are shown in Figure 5 for a full day of incubation; concentrations of clin-MIP polyurethane nanofiber up to 450 µg/mL were incubated without producing any unfavorable side effects. More than 90% of the cells were living cells. Thus, it can be concluded that the Clin-MIP polyurethane nanofiber solution showed no negative effects at any of the evaluated concentrations. Clindamycin MIP is included in polyurethane nanofibers to provide controlled release of the medication over a longer duration. Clindamycin’s cytotoxic effects on cells are reduced by this controlled release profile, which aids in maintaining therapeutic concentrations of the drug. The remarkable cell survivability may have been further attributed to optimizing the nanofiber construction technique, which guaranteed consistent medication distribution and minimum damage to encapsulated cells during manufacturing. The high cell viability (>85%) seen in the cytotoxicity assessment of the clindamycin MIP-loaded polyurethane nanofibers is a result of a combination of biocompatible materials, molecular imprinting technology, regulated drug release, and an optimized production process [40]. This indicates the formulation’s potential in biomedical applications requiring minimum cytotoxicity and sustained drug delivery, such as in tissue engineering clindamycin delivery systems or the localized treatment of acne infections. This finding confirms that the polyurethane nanofibers loaded with clindamycin MIP are biocompatible with skin tissue, which is not unexpected considering that the formulation’s constituents are generally considered safe [39].

### 3.8. Determination of Antimicrobial Activity

#### 3.8.1. Antimicrobial Activity of Drug Formulation

The results of the antimicrobial activity evaluation revealed that the Clin-MIP polyurethane nanofiber has potential activity against all tested strains compared with blank (Clin.) and control (Gentamycin). Large inhibition zones ranging from 24 mm to 45 mm were determined by the tested formula against *S. epidermidis*, *S. mutans*, and *S. aureus*. Although the blank has no activity against the tested Gram-negative strains, the new clindamycin nanofiber showed good antimicrobial activity against *K. pneumoniae*, *P. vulgaris*, and *P. aeruginosa,* with inhibition zones larger than those of Gentamycin (Figure 6a,b). 

#### 3.8.2. Anti-Proliferation Activity of Clin. Nano Formula

The MIC and MBC values confirmed the Clin-MIP polyurethane nanofiber’s higher activity against the tested organisms than Clin. The Clin-MIP polyurethane nanofiber has the lowest MICs and MBC values of all tested strains. In contrast, blank (Clin) has antibacterial activity against two strains, *S. aureus* and *S. mutans*, with higher MIC and MBC values (Table 4). The Clin-MIP polyurethane nanofiber exhibited the highest antibacterial activity against the *S. aureus* strain, with MIC and MBC values of 0.39 μg/mL and 6.25 μg/mL, respectively. 

#### 3.8.3. Bacterial Killing Kinetic Assay

The bacterial killing kinetics of the Clin-MIP polyurethane nanofiber against *S. aureus* (ATCC 6538) were assessed and compared with Clin to better investigate its killing capacity. After 180 min of incubation, Clin-MIP polyurethane nanofiber (1× MIC) demonstrated strong bactericidal activity, as illustrated in Figure 7. Clindamycin (1× MIC) demonstrated a slower killing kinetic under the same conditions, although it could still not eradicate all bacteria after 180 min.

### 3.9. In Vivo Study 

#### 3.9.1. Assessment of Ear Thickness and Anti-Inflammatory Potential

The visual examination of the rat ears in Figure 8 compares the left ear side (untreated group) and the right ear side treated group; inflammation was evident in all groups receiving Staphylococcus aureus treatment, as evidenced by a noticeable increase in ear thickness and redness. As seen in Figure 8, groups II and III experienced a notable decrease in inflammation symptoms. Furthermore, it was discovered that group III (treated with Clin-MIP polyurethane nanofiber) had a very high reduction in inflammation (ear thickness). In contrast, group II (treated with Clin-MIP) had a high reduction, and group I (treated with Clin-Suspension) had a low reduction. Preclinical research conducted in a rat model of acne vulgaris by Tolentino et al., 2021 consistently supported these findings by demonstrating that clindamycin may have potent antibacterial and anti-inflammatory targets using polymeric nanocarriers in the event of oily skin disorders; this targeting was enhanced [7].

The immunomodulatory effects of clindamycin were linked to decreased TNF-α production and NF-κB suppression. These effects were also seen in a clindamycin derivative that helped treat painful and inflammatory diseases but had less antibacterial activity; this is supported by Rodrigues et al., 2023 [41].

Molecular imprinting creates polymer matrix cavities that match the target molecule’s size, shape, and functional groups. Polyurethane nanofibers molecularly imprint clindamycin to capture and release it slowly, improving its action at the application site. Adding clindamycin to polyurethane nanofibers allows for continuous medication release. This delayed-release approach maintains effective clindamycin concentrations at the application site, prolonging its benefits for acne bacteria and inflammation. Polyurethane nanofibers are molecularly imprinted to carry clindamycin to the target spot. Concentrating clindamycin at the site of infection and inflammation with molecularly imprinted polyurethane nanofibers can improve acne treatment while reducing systemic exposure and side effects.

Clin-MIP Polyurethane nanofiber has a potential effect for treating acne and inflammation by increasing the stratum corneum penetration of the Clin-MIP Polyurethane nanofiber combination, notably when integrated into molecularly imprinted polyurethane nanofibers, mainly inhibits bacterial protein production. It also inhibits peptide bond formation and bacterial development by binding to the 50S ribosomal subunit of sensitive bacteria such as Propionibacterium acnes. Clindamycin reduces comedones and inflammatory acne lesions by lowering the amount of P. acne on the skin. Clin blocks pro-inflammatory cytokines, including IL-6 and TNF-alpha, which cause acne inflammation. Clindamycin reduces acne lesion redness, edema, and discomfort by lowering inflammation.

#### 3.9.2. Healing Activities of Clin-MIP Polyurethane Nanofiber Using the Viable Count Method in an Animal Model

The in vivo antimicrobial activity of clindamycin and Clin-MIP polyurethane nanofiber significantly decreased the number of *S. aureus* colonized in infected rat ears compared to the control (untreated model) that was only injected by S. aureus. After completing a treatment period, the bacterial counts of *S. aureus* dropped from 1 × 10^8^ to 39 × 10^1^ CFU/mL and 89 × 10^1^ CFU/mL using both Clin-MIP and Clin-MIP polyurethane nanofiber, respectively. In contrast, the bacterial number of the negative control (untreated) was slightly increased to 43 × 10^8^ CFU/mL. In contrast, the positive control treated with clindamycin solution reduced the bacterial number to 92 × 10^2^ CFU/mL, as shown in Figure 9.

#### 3.9.3. Assessment of Inflammatory Biomarkers

Impact of Clin suspension, Clin-MIP, and Clin-MIP polyurethane nanofiber therapy on the protein profiles of serum inflammatory cytokines

LPS alters the inflammatory cytokines that cause skin inflammation associated with acne. In earlier research, we found that administering Clin suspension, Clin-MIP, and Clin-MIP polyurethane nanofibers to inflammation decreased the amounts of NLRP3, TNF-α, IL-1β, and IL-6 generated by LPS in vitro. Three hours after delivering LPS to rats, we collected serum samples to examine the impact of Clin suspension, Clin-MIP, and Clin-MIP polyurethane nanofiber on serum cytokine profiles. Clin-MIP polyurethane nanofiber therapy dramatically decreased the expression of NLRP3, TNF-α, IL-1β, and IL-6 (40.39%, 79.6%, 70.50, and 73.30%, respectively) in comparison to the LPS group (Figure 10). One important part of the innate immune system is NLRP3, and pro-inflammatory cytokines are produced when this protein is activated. Lower levels of NLRP3 suggest that this inflammatory pathway is inhibited, which reduces the generation of cytokines like IL-1β and IL-16. IL-1β, a well-known pro-inflammatory mediator in acute inflammation, indicates that tumor inflammation may benefit from Clin-MIP polyurethane nanofibers. Numerous investigations have shown that immunoregulatory cytokines modify the inflammation caused by wounds. TNF-α regulates IL-6 levels during the initial stages of inflammation. The alarm-phase cytokine IL-1β has been connected to several symptoms associated with wounds. Figure 10 shows increased levels of IL-1β receptor antagonists and TNF-α suppression. Inflammation is a major component of skin pathology. Adipose lipolysis is triggered by skin tissue leakage and produces toxic unsaturated fatty acids, such as arachidonic acid, a building block of pro-inflammatory eicosanoids. They produce excessive inflammatory markers, which can exacerbate illness and cause blisters. Pathogenesis and illness are encouraged by the innate immune system. Three hours after LPS injection, our research showed that treating Clin with Clin-MIP polyurethane nanofiber decreased TNF-α, IL-1β, and IL-6 blood levels. Clin-MIP polyurethane nanofiber may reduce the inflammatory response to acne skin inflammation by modulating immunoregulatory cytokines [1,42]

### 3.10. Histopathological Examination

The histological results for the group treated with an alternative formulation on the right ear side showed fewer abnormalities than the left ear side, as shown in Table 5 and Figure 11. Figure 11a illustrates Group I, the left ear side (untreated group serving as control), which exhibited considerable epidermal cell hypertrophy, an apparent increase in epidermal cell quantity (hyperplasia), and a moderate degree of thick stratum corneum covering (hyperkeratosis). Moderately thick coating of keratin. The dermal layer has a significant rate of inflammatory cell invasion. In the dermis, there was severe congestion. Certain sebaceous glands are destroyed. Clin-Suspension is used on the right ear side, and a few mildly inflammatory cells are in the dermis. There were many hair follicles with sebaceous glands in the dermis. Most of the epidermal layers had a normal keratin layer covering them with normal thickness. Most dermal layers showed a normal ordering and were free of congestion.

Figure 11b: This depicts Group II, the left ear side (untreated group), where numerous areas of the epidermis are covered in a somewhat thick keratin coating. There is a thick spot in the epidermis. The dermis showed significant inflammatory cell infiltration, and the dermal layer showed congestion. The dermis under the skin on the right ear side (treated with Clin-MIP) shows a small degree of congestion and a tiny infiltration of inflammatory cells. Most hair follicles were connected to sebaceous glands and were normal. The standard thickness was disclosed nearly to the epidermal layer. There was no hyperplasia noted. A thick coating of keratin covers a few regions of the epidermis.

Figure 11c: demonstrates Group III, the left side (untreated group), where the thickness of the keratin layer has significantly (but not significantly) grown. Significant inflammatory cell infiltration exists in the underlying dermis, and the dermis layer showed significant congestion and disarray. There was mild hyperplasia of epidermal cells on the right side (treated with Clin-MIP polyurethane nanofiber), with certain areas exhibiting normal epidermal thickness. A slight degree of inflammatory cell infiltration occurs in the dermis. Numerous hair follicles were visible in several places. In the dermis, no congestion was observed.

Following treatment with three distinct formulae, the skin’s hyperkeratosis, inflammatory cell infiltration, epidermal hypertrophy, and congestion improved. Clin-MIP was administered to Group II and Group III, respectively. Compared to Group I, which received clin-suspension treatment, polyurethane nanofibers recuperated more quickly. Compared to Group II, Group III is better. Group III is the most effective formula in comparison to other formulas. This implies that, after being used to cure an infection or skin disease, the mixture may help the skin layers.

## 4. Conclusions

This work involved synthesizing and characterizing mixed molecular imprinting polymer-rated into polyurethane nanofiber as a nanocarrier for topical clindamycin administration to increase clindamycin antibacterial activity. Combination therapy was thought to be a new tactic for combating bacterial resistance and severe side effects (Graphical Abstract). Clin-MIP polyurethane nanofibers with homogenous morphology and floral forms may be successfully synthesized without interpolating between different polymers. Examined in rats’ ears, earlobes coated with clin-MIP polyurethane nanofibers revealed a 1.3-fold increase in permeability and little signs of inflammation, such as redness and papules. The study suggests that co-delivering clindamycin together with mixed MIP and nanofiber may enhance clindamycin effects at the administration site and reduce the risk of antibiotic resistance emerging.

## Figures and Tables

**Figure 1 pharmaceutics-16-00947-f001:**
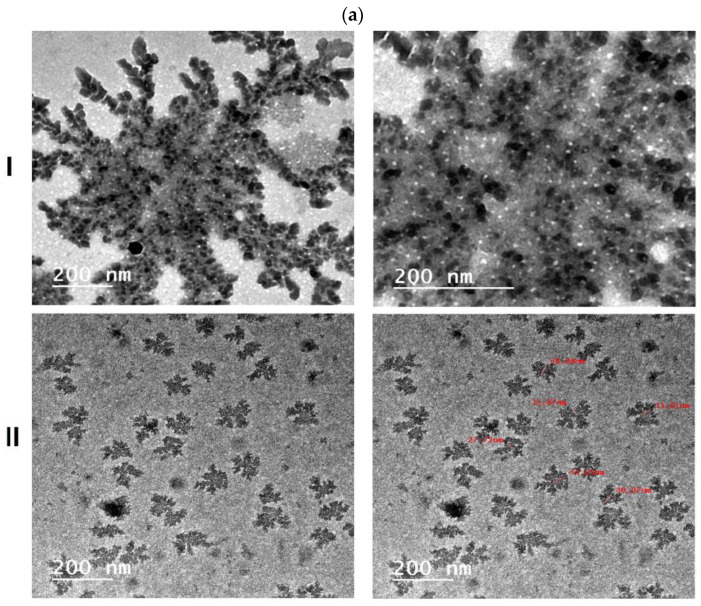
(**a**): Transmission Electron Microscopy (TEM) imaging of (**I**) molecularly imprinted polymer (Clin-MIP) and (**II**) Non-molecularly imprinted polymer without clindamycin (NIP); (**b**): (**I**) Fourier Transformer Infrared Spectroscopy (FTIR) (**II**) Discussion of Differential Scanning Calorimetry for Non-imprinting polymer (NIP), Clindamycin (Clin), and Clindamycin molecularly imprinted polymer (Clin-MIP), respectively. (**c**): Adsorption isotherms of (**I**) Clindamycin (Clin) and (**II**) Lincomycin (Lin) on imprinted and non-imprinted particles and their chemical structures. (**d**) Scatchard analysis. (**e**) Adsorption kinetic curves for Clin-MIP and Non-Imprinted Polymer (Clin-NIP).

**Figure 2 pharmaceutics-16-00947-f002:**
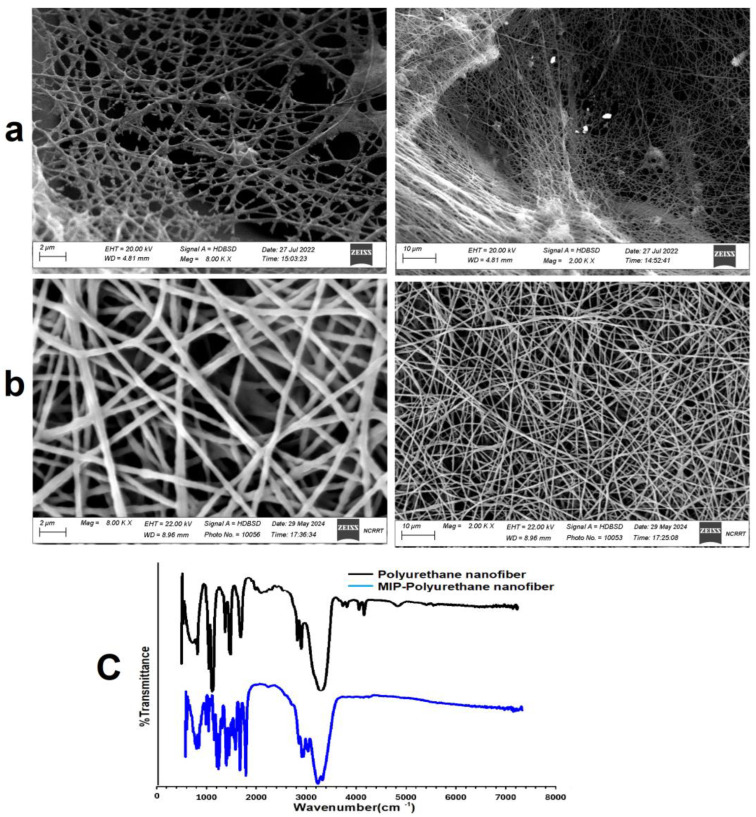
SEM images of silk (**a**) Clin-MIP polyurethane nanofiber (CMPN), (**b**) Polyurethane nanofiber (PN), and FTIR of PN and CMPN, (**c**) Polyurethane nanofibers’ FTIR spectra.

**Figure 3 pharmaceutics-16-00947-f003:**
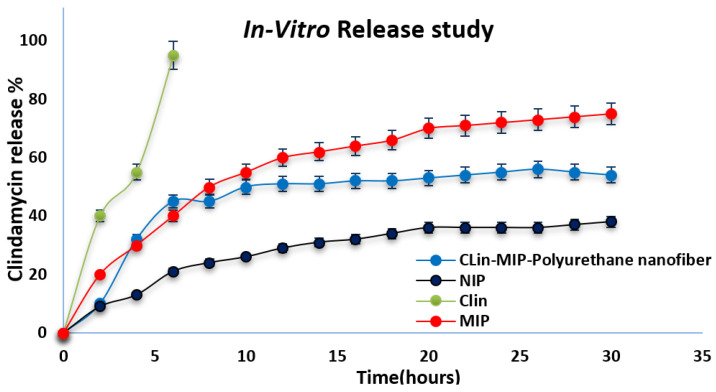
Release profiles of clindamycin from Clin-MIP and Clin-MIP polyurethane nanofibers compared to a clindamycin solution.

**Figure 4 pharmaceutics-16-00947-f004:**
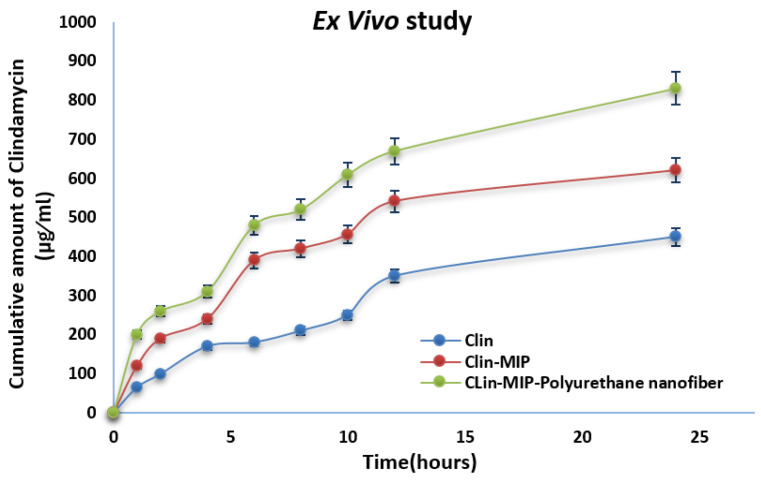
Ex-vivo skin penetration of clindamycin from Clin-MIP and Clin-MIP polyurethane nanofiber compared to a clindamycin solution.

**Figure 5 pharmaceutics-16-00947-f005:**
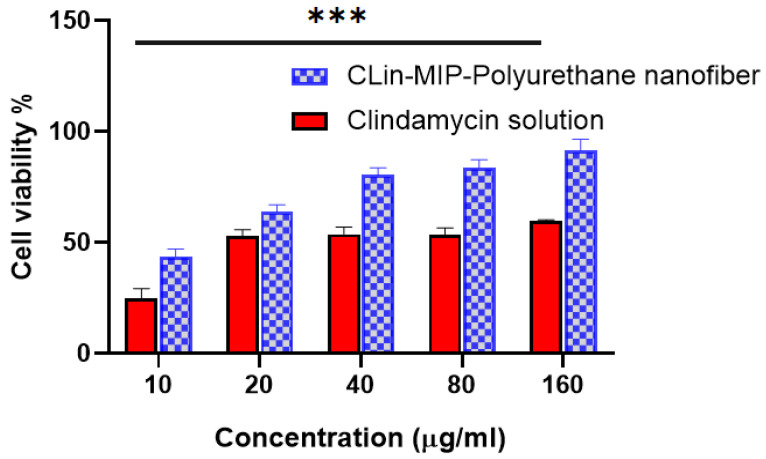
Clindamycin loaded with MIP polyurethane nanofiber affects HSE-2 cell viability. 100% cell viability and an untreated cell’s average MTT reduction value correlate. *** Mean *p* < 0.0015 for ANOVA.

**Figure 6 pharmaceutics-16-00947-f006:**
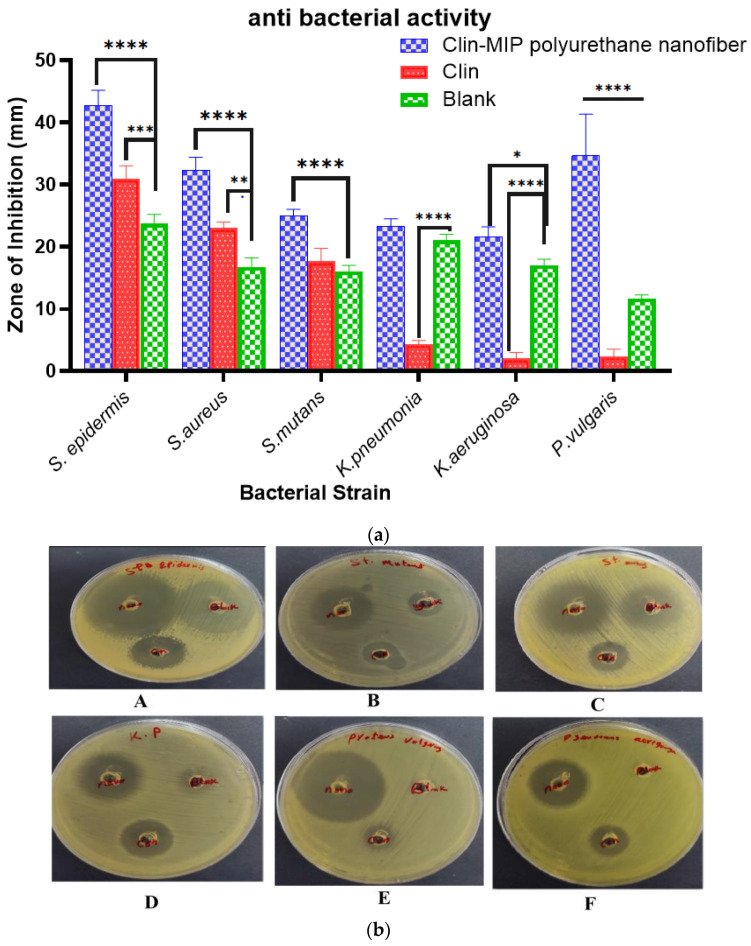
(**a**). Antibacterial activity of Clindamycin and Clin-MIP polyurethane nanofibers against skin profile bacterial strains and (**b**). Antibacterial activity of Clin. and Clin-MIP polyurethane nanofibers against *S. Epidermis* (**A**), *S. mutant* (**B**), *S. aureus* (**C**), *Pneumonia* (**D**), *P. Vulgaris* (**E**), and *P. argues* (**F**). Blank: Clindamycin-Control: Gentamycin. Note: * *p* = 0.0189, ** *p* = 0.0015, *** *p* = 0.0003 and **** *p* < 0.0001.

**Figure 7 pharmaceutics-16-00947-f007:**
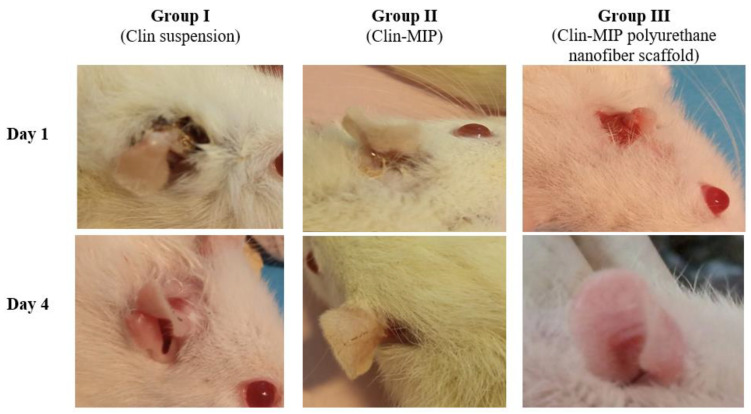
Killing kinetics of Clindamycin and Clin-MIP polyurethane nanofiber at 1× MIC of concentration against *S. aureus* (ATCC 6538) for different time intervals.

**Figure 8 pharmaceutics-16-00947-f008:**
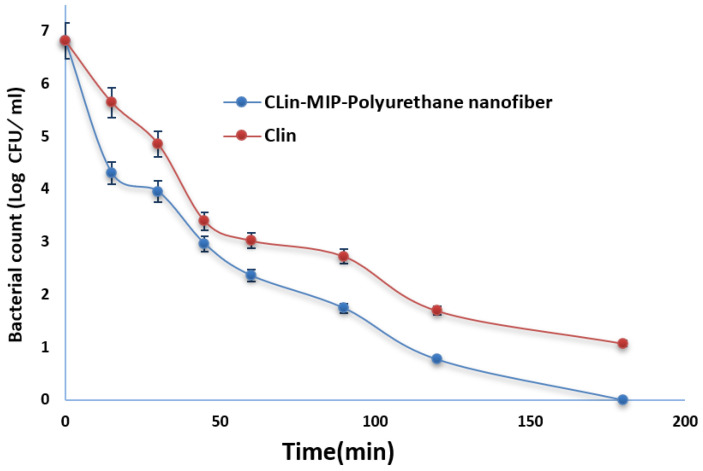
Morphological alterations in a rat ear model to identify possible anti-inflammatory.

**Figure 9 pharmaceutics-16-00947-f009:**
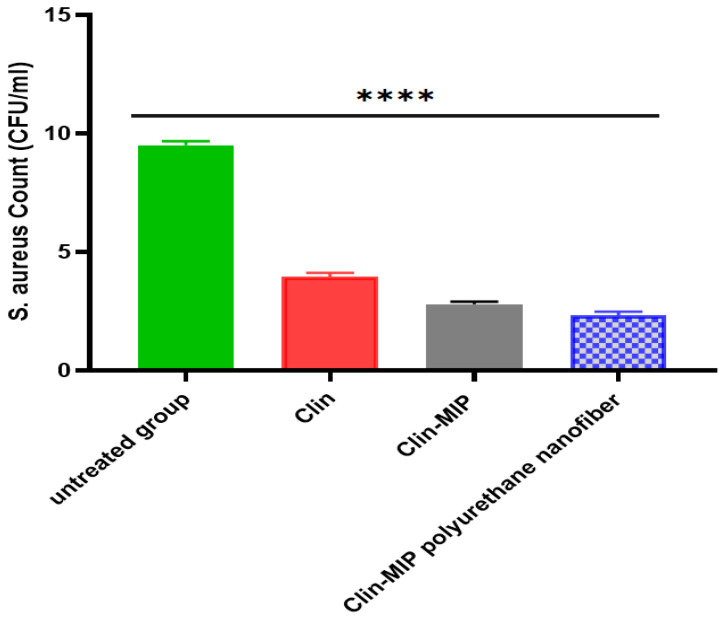
The total number of *S. aureus* (CFU/mL) recovered from rats’ ears after treatment with Clin-MIP polyurethane nanofiber. **** *p* < 0.0001.

**Figure 10 pharmaceutics-16-00947-f010:**
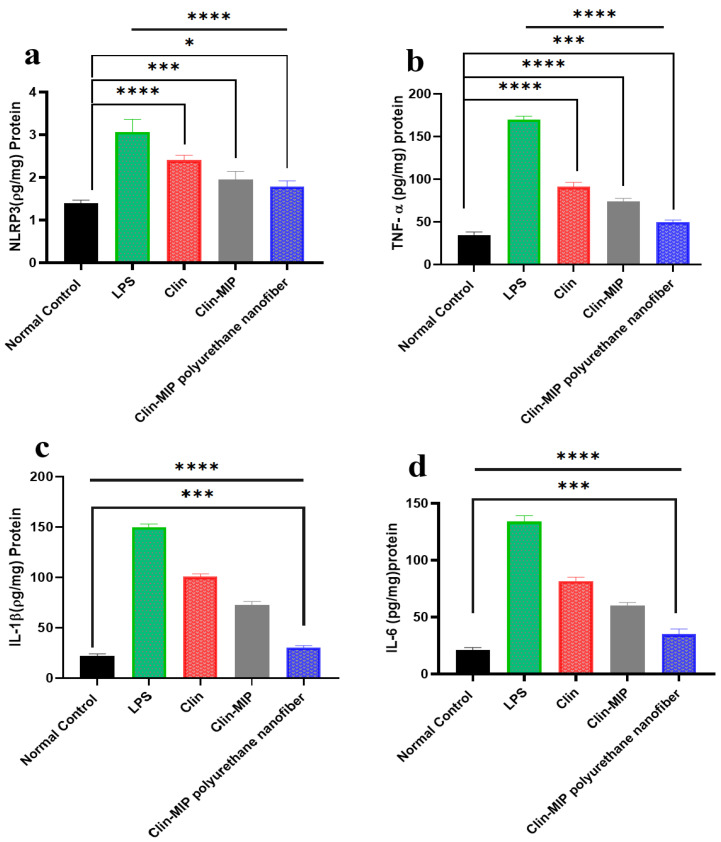
The effect of Clin suspension, Clin-MIP, and Clin-MIP polyurethane nanofiber on the inflammatory-related markers NLRP3, TNF-α, IL-1β, and IL-6. Note: * *p* = 0.0223, *** *p* = 0.0008 and **** *p* < 0.0001.

**Figure 11 pharmaceutics-16-00947-f011:**
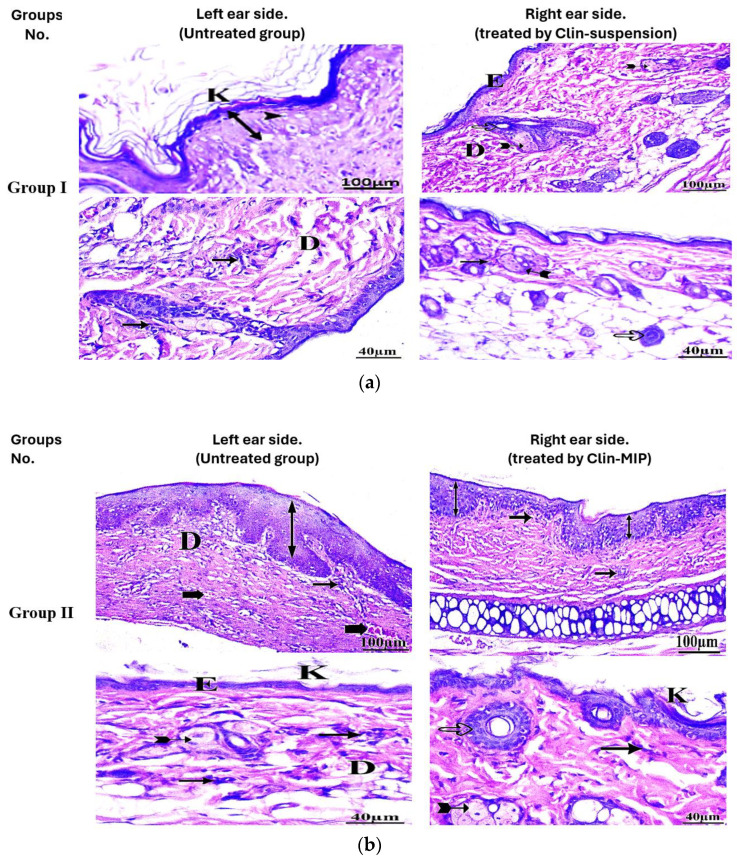
(**a**) Displays photomicrographs of skin specimens from Group 1. On the left side, the images reveal interrupted dermis, congestion, vacuolated epithelial cells, hyperkeratosis, and cellular infiltration in the dermis. Increased cellular proliferation of the epidermis and disrupted hair follicles are also evident. On the right side, treated with Clin-Suspension, the epidermis and dermis appear intact, with congestion, hyperkeratosis, cellular infiltration, epithelial hyperplasia, and intact sebaceous glands. In (**b**), the untreated Group II exhibits disordered dermis, congestion, vacuolated epithelial cells, hyperkeratosis, and cellular infiltration in the dermis. Similar to Group 1, increased cellular proliferation of the epidermis and disrupted hair follicles are observed. In contrast, Group II treated with Clin-MIP shows improvement, with the entire hair follicle visible, minimal congestion, and intact sebaceous glands. (**c**) showcases skin specimens from Group III. On the left side, congestion, vacuolated epithelial cells, hyperkeratosis, and cellular infiltration in the dermis are evident, along with increased cellular proliferation of the epidermis. However, on the right side treated with Clin-MIP-Polyurethane nanofiber, the epidermis and dermis appear healthier, with visible hair follicles, minimal congestion, and intact sebaceous glands. Note: Epidermis (E), dermis (D), and Hyperkeratosis (K).

**Table 1 pharmaceutics-16-00947-t001:** (**a**). Formulations, particle size, and polydispersity index Yield of Clin-Imprinted and Non-Clin Imprinted Polymers. (**b**). The percent of Clindamycin (Clin) and Lincomycin (Lin) that are bound by imprinted (MIP) and non-imprinted (NIP) particles, as well as the α and ε values for various Ci concentrations. The data are presented as the mean ± S.D. (**c**). Values of K_a_, B_max_, and R^2^ by Scatchard analysis. (**d**). Kinetics data for Clin-MIP and Clin-NIP.

**a**									
**Sample**	**Clindamycin** **Template (mg)**	**MAA** **(mmol)**	**AIBN** **(mg)**	**EGDMA** **(mmol)**	**Particle Size** **(nm)**		**Polydispersity Index** **(PdI)**		
NIP 1	0	2	150	10	84		0.56		
NIP 2	0	4	150	20	45		0.51		
MIP 1	424.98	2	150	10	150		0.44		
MIP2	424.98	4	150	20	110		0.49		
**b**									
**Ci (mol/L)**	**Bound Clin (%)**		**Bound Lin (%)**		**α Clin**		**α Lin**		**ε**
	**MIP**	**NIP**	**MIP**	**NIP**					
0.25	48 ± 0.31	39 ± 0.61	18 ± 0.30	17 ± 0.60	1.23		1.06		2.67
0.5	59 ± 0.61	43 ± 0.31	20 ± 0.60	18 ± 0.30	1.37		1.11		2.95
0.75	67 ± 0.21	49 ± 0.12	24 ± 0.20	21 ± 0.10	1.37		1.14		2.79
1	79 ± 0.21	51 ± 0.01	26 ± 0.20	22 ± 0.01	1.55		1.18		3.04
1.25	78 ± 0.21	57 ± 0.01	31 ± 0.20	29 ± 0.01	1.37		1.07		2.52
**c**									
**Polymer**	**High-Affinity Sites**			**Low-Affinity Sites**					
	**K_a_ (M^−1^)**	**B_max_ (mM/g)**	**R^2^**	**K_a_ (M^−1^)**		**B_max_ (mM/g)**		**R^2^**	
MIP	0.7347	0.1123	0.933	0.0891		0.1642		0.8845	
NIP				0.1469		0.0197		0.9495	
**d**									
**Polymer**	**Pseudo-First Order**			**Pseudo-Second Order**					
	**Qe** **(mol/g)**	**K_1_** **(h^−1^)**	**R^2^**	**Qe ** **(g/mol/h)**		**K_2_** **(h^−1^)**		**R^2^**	
MIP	0.0003286	0.1309	0.9994	−3.718 × 10^−6^		6.42 × 10^11^		0.9842	
NIP	0.0007199	0.0181	0.9995	4.06 × 10^9^		−0.000806		0.9885	

**Table 2 pharmaceutics-16-00947-t002:** Physicochemical characteristics of nanofibers (mean ± SD).

Formulation	Weight Uniformity (mg)	Thickness (mm)	Folding Endurance (Times)	Tensile Strength (MPa)
Clin-MIP-polyurethane nanofiber	31.54 ± 0.1	0.28 ± 0.01	174 ± 3	1.82 ± 0.03
polyurethane nanofiber	22.65 ± 0.2	0.18 ± 0.01	154 ± 5	1.99 ± 0.02

**Table 3 pharmaceutics-16-00947-t003:** Ex-vivo pharmacokinetic parameters of clindamycin from Clin-MIP and Clin-MIP polyurethane nanofibers compared to a clindamycin solution estimated by linear regression.

Formula	Jss (µg/cm^2^/h)	Kp (cm/h)	Q24 (µg/cm^2^)	Skin Deposition (µg/cm^2^)
Clin suspension	15 ± 0.21	1.5 ± 0.11	450 ± 0.43	123.75 ± 0.64
Clin-MIP	12.78 ± 0.32	1.28 ± 0.42	620 ± 0.45	169.75 ± 0.55
Clin-MIP polyurethane nanofiber	19.44 ± 0.33	0.6075 ± 0.41	830 ± 0.13	171.015 ± 0.23

**Table 4 pharmaceutics-16-00947-t004:** Minimum inhibitory concentrations (MIC) and minimum bactericidal concentrations (MBC) of clindamycin and Clin-MIP polyurethane nanofiber against the skin profile-tested bacterial strains.

Bacterial Strains	Clindamycin (Blank)		Clin-MIP Polyurethane Nanofiber	
	MIC (μg/mL)	MBC (μg/mL)	MIC (μg/mL)	MBC (μg/mL)
*Staphylococcus epidermidis* (ATCC 12228)	12.5 ± 0.2	12.5	6.25 ± 1.8	6.25
*Staphylococcus aureus* (ATCC 6538)	3.125 ± 0.9	6.25	0.39 ± 0.2	0.39
*Streptococcus mutans* ATCC 25175	3.125 ± 0.9	12.5	0.78 ± 1.3	3.125
*Klebsiella pneumoniae* (ATCC13883)	NA	NA	1.56 ± 0.9	3.9
*Pseudomonas aeruginosa* (ATCC90274)	NA	NA	6.25 ± 0	6.25
*Proteus vulgaris* (ATCC13315)	NA	NA	1.56 ± 0.45	3.9

NA: no activity.

**Table 5 pharmaceutics-16-00947-t005:** Shows the grade of histopathological findings on ear rat skin.

Groups	Left F1			Right F1			Left F2			Right F2			Left F3			Right F3		
Number of examined fields	10			10			10			10			10			10		
Grade	-	+	-	+	-	+	-	+	++	-	+	++	-	+	++	-	+	++
Abnormal changes																		
Hyperkeratosis	5	0	5	6	3	1	2	5	3	4	4	2	3	5	2	7	3	0
Inflammatory cells infiltration	2	6	2	4	6	0	1	5	4	2	8	0	1	7	2	4	6	0
Thick epidermis (hypertrophy)	6	1	3	9	1	0	4	3	3	8	2	0	4	1	5	9	1	0
Dermal congestion	5	3	2	8	2	0	4	3	3	7	3	0	4	3	3	9	1	0

Grade of histopathological findings: - (no abnormality), + (slight), ++ (mild).

## Data Availability

Data is available on request from the authors.

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
