# Peer review of "Enhanced Antibacterial Activity of Clindamycin Using Molecularly Imprinted Polymer Nanoparticles Loaded with Polyurethane Nanofibrous Scaffolds for the Treatment of Acne Vulgaris"

_pharmaceutics, 2024, doi:10.3390/pharmaceutics16070947_

Round 1
Reviewer 1 Report
Comments and Suggestions for Authors
The authors developed clindamycin molecular imprinting polymeric nanoparticles loaded onto polyurethane nanofiber scaffolds for acne treatment. The main idea of this work is good and important but it requires a big improvement and additional investigations. Therefore, I address a major revision of the paper.
1. A lot of grammar mistakes in the manuscript such as the sentence in line 33-34: “Methylacrylic acid (MAA), Ethylene glycol dimethacrylate (EGDMA), and azoisobutyronitrile (AIBN)as functional monomer, crosslinker, and radical initiator.”, line 189 “Imprinted Molecularly nanoparticles”
2. Reformulate the sentences in lines 35-38
3. Check some repeated abbreviations such as MIP in line 173. Sometimes, the authors use the full name while they have already used the abbreviation before. For example, Molecularly imprinted polymer in line 180, MAA, EGDMA and AIBN in lines 381-382, etc. Check all the manuscript.
4. The authors should provide more details about the clindamycin-containing polymer powder. Is it the empty MIP followed by the adsorption of clindamycin. If so, please provide the conditions of adsorption “solvent, concentration, weight of the polymer, adsorption time, etc…
5. In line 191, “14 weight percent polyurethane” If the authors mean 14% of polyurethane and the rest if the MIP powder, I think that 14 % of polyurethane is very low leading to fiber with very low mechanical properties. Did the authors did any optimization?
6. What do the authors mean by “70 parts tetrahydrofuran”? Specify the amount of the polymer and volume of the solvent.
7. The authors should specify which polymer they are talking about specially in 192, 194, etc since two polymers are used including polyurethane and MIP.
8. The term “mixed” in line 193 should be changed by “dispersed” or other.
9. In lines 193-194, the authors should specify the volume of Clin-MIP solution and the polyurethane solution and the ration between the two solution. Is it 1:1 or 1:2 or 2:1 or what ?
10. I could not understand the lines 205-206.
11. I encourage the authors to draw some schematic illustrations in the section 2.3 covering the preparation of the MIP, the preparation of the MIP-nanofibers via electrospinning. Moreover, one other scheme to understand the Ex Vivo Skin Permeation Study.
12. In lines 382-383, add the monomer because MAA is a monomer not a crosslinker and try to avoid using plural words ‘crosslinker” not crosslinkers and so on.
13. In the section 3, the authors must add the results on the adsorption capacity of MIP and NIP at different concentrations and calculate the imprinting factor.
14. In Figure 1b, the authors should delete “Clin and” and identify the main peaks in the spectra. Moreover, They need to add the spectrum of empty MIP after the removal of the template. Also, the spectrum of the MIP-nanofibers and nanofibers without MIP.
15. The authors should add the scale in the figure 2. It would be good if the authors add the SEM of the polyurethane fibers without MIP.
16. In figure 3, the authors should add the release profile of the NIP-nanofiber also.
17. In Figure 3, all the samples must contain the same number of moles of Clin. How did the authors ensure that? Please specify the release conditions in the figure caption such as number of moles of Clin, etc. Similar comment for Figure 4.
18. Add the results of the Clin-NIP polyurethane nanofiber in Table 2
Comments on the Quality of English Language
The english is somewhat good but various mistakes are detected
Author Response
Dear reviewer,
thank you so much for considering our work, Your expertise is highly appreciated. we carefully considered all raised comments as listed in the following points and responses. in the revised version of the manuscript, all corrections were highlighted in yellow. we seriously wish that the modifications made in the manuscript are satisfactory.

Reviewer 2 Report
Comments and Suggestions for Authors
The paper describes an interesting approach and should be published after minor additions.
The authors describe in “Methods”
“The template was removed using a Soxhlet apparatus….” However no data about rebinding, esp. the concentration dependence leading to the Kd-value, are given These measurements are necessary for characterizing the performance of the MIP.
Furthermore it is not clear whether the MIP after the synthesis without removal of the template or after reloading of the MIP was integrated in the scaffold.
Author Response
Dear reviewer,
Thank you so much for considering our work. We appreciate your expertise. We carefully considered all comments raised, as listed in the following points and responses in the revised version of the manuscript. All corrections were highlighted in yellow. We earnestly wish the modifications made in the manuscript were satisfactory.

Reviewer 3 Report
Comments and Suggestions for Authors
In this manuscript, the authors present the preparation of a clindamycin-MIP-based material for the treatment of acne. Although the authors have successfully prepared the material, the novelty of the study is lacking. The study requires more experimentation, especially when it comes to the contribution of MIP to release kinetics. The results obtained are not thoroughly presented and sufficiently explained. NIP controls for the in vitro characterisation of Clin-MIP are missing. The manuscript requires further experimentation and discussion for further consideration in Pharmaceutics.
1) The objectives and rationale of the study are clearly stated, but are not thoroughly validated with experiments; comparison with NIP controls, especially in the in vitro release study.
2) Section 2.3.1.: What is the final concentration of clindamycin and MAA in the pre-polymer mixture
3) Section 2.3.4.5: How many replicates were performed?
4) MIP perse is a polymer with specific design cavaties in the matrix; TEM analysis should also be performed for MIP without clindamycin and MIP.
5) Adequate statistical analysis was not performed in many of the figures. Please improve the statistical reporting.
6) Line 71: Authors should put the abbreviation “sp.” or “spp.” after the genus name if they mean species or species plura.
7) Incorrect use of capital letters in the text; names of antibiotics (unless they are brand names), molecularly imprinted polymers, comedones are all written in lower case. I ask the authors to use the correct use of initials throughout the text.
8) Lines 54-55: Pityrosporum, Cutibacterium granulosum and C. acne should be italicised. Also, in line 55, the bacterium acne should be written with its full name.
Author Response
Dear reviewer,
Thank you so much for considering our work. Your expertise is highly appreciated. We carefully considered all comments listed in the following points and responses. In the revised version of the manuscript, all corrections were highlighted in yellow. We earnestly hope that the modifications made to the manuscript are satisfactory.

Round 2
Reviewer 1 Report
Comments and Suggestions for Authors
The authors improve the manuscript. I address these few comments to authors.
In lines 183-184, we should not say MIP polymer, but MIP. The same comment for line 192.
Add the ticks in the two FTIR figures on wavenumber axis and delete the values in the Tansmittance axis. Add ticks on x-axis also in the Figure of DSC. Please do that for all figures.
In figure 7, please separate the two images of Group II (Day 1 - Day 4).
The authors should revise the numbering of figures.
Author Response
Thank you so much for your notes and comments

Reviewer 3 Report
Comments and Suggestions for Authors
The authors have mostly considered and incorporated my comments, but there are still a few things I recommend to take into account before publishing the manuscript in Pharmaceutics.
1. I understand that the authors use in vitro studies to interpret the kinetics of release. However, the purpose of MIP is to allow strong binding of the drug due to the specific binding sites of the imprinted cavities. In contrast, with NIP, the binding of the drug occurs through interactions at the surface of the matrix, resulting in faster diffusion of the drug. On this basis, I would also recommend a comparison with NIP in an in vitro release study to show that it is indeed the specific binding of clindamycin in the MIP.
2. Section 2.3.4.5: How many replicates were performed in vitro drug release study?
3. Lines 58, 59: name of orders are written with capital letters
4. Line 200: weight % should be written wt.% or wt%.
5. Throughout the manuscript there are still incorrect use of capital letters (e.g., in lines 176, 312).
Author Response

(The authors gave the same response as above.)

Round 3
Reviewer 3 Report
Comments and Suggestions for Authors
The authors sufficiently addressed the remaining comments.